# NRF2 deficiency increases obesity susceptibility in a mouse menopausal model

Xunwei Wu[1,2], Jun Huang[3], Cong Shen[4], Yeling Liu[4], Shengjie He[4], Junquan Sun[4], Bolan Yu[1,2]*

**1** Key Laboratory for Major Obstetric Diseases of Guangdong Province, The Third Affiliated Hospital of Guangzhou Medical University, Guangzhou, China, **2** Guangdong Engineering and Technology Research Center of Maternal-Fetal Medicine, The Third Affiliated Hospital of Guangzhou Medical University, Guangzhou, China, **3** Department of Obstetrics, The Third Affiliated Hospital of Guangzhou Medical University, Guangzhou, China, **4** Third Clinical College, Guangzhou Medical University, Guangzhou, China

* yubolan-q@qq.com

**Data Availability Statement:** All relevant data are within the paper and its Supporting Information files.

**Funding:** This study was supported by the Key Project of Guangzhou Science and Technology

## Abstract

The risk of metabolic abnormalities in menopausal women increases significantly due to the decline in estrogen level. Nuclear factor E2-related factor 2 (NRF2) is an important oxidative stress sensor that plays regulatory role in energy metabolism. Therefore, an ovariectomized menopausal model in *Nrf2*-knockout (KO) mice was applied to evaluate the effect of *Nrf2* deficiency on metabolism in menopausal females. The mice were divided into four groups according to their genotypes and treatments. Blood samples and bodyweights were obtained preoperatively and in the first to ninth postoperative weeks after overnight fasting. Serum levels of triglycerides (TG), total cholesterol (T-CHO), low-density lipoprotein (LDL), high-density lipoprotein (HDL), and glucose (GLU) were measured at postoperative weeks 0, 1, 3, 5, 7, and 9. Neurotransmitter dopamine (DA) and serotonin (5-HT) was analyzed in brain tissues after sacrifice at postoperative week 9. The results demonstrated that, compared with the corresponding wild-type (WT) mice, KO ovariectomized mice had a greater bodyweight gain (*P*<0.01). Serum analysis showed that the serum GLU, T-CHO, and TG were significantly lower (*P*<0.05) but LDL was significantly higher (*P*<0.05) in the KO control mice than that in WT control mice. However, different from the WT counterparts, an increase in blood GLU level (*P*<0.05), unchanged T-CHO, TG, and HDL levels, and a significant reduction in LDL (*P*<0.01) was found in the KO ovariectomized mice. In addition, the level of 5-HT was significantly reduced (*P*<0.05) in the KO mice after ovariectomy. In conclusion, the combination of *Nrf2* deletion and a decline in estrogen level induced a significant increase in bodyweight, which may be associated with their altered glucose and LDL metabolism and decreased 5-HT levels. From a clinical perspective, women with antioxidant defense deficiency may have an increased risk of metabolic abnormalities after menopause.

Innovation Committee (grant number 201804020057), Guangdong Science and Technology Department Project (grant number 2019B030316023), and Lin He's Academician Workstation of New Medicine and Clinical Translation at the Third Affiliated Hospital. BY received all the funding. The funders had no role in study design, data collection and analysis, decision to publish, or preparation of the manuscript.

**Competing interests:** The authors have declared that no competing interests exist.

## Introduction

Menopause refers to the cessation of ovarian function that leads to the termination of the menstrual cycle and is accompanied by decreased estrogen levels, increased follicle stimulating hormone and luteinizing hormone levels, and significant changes in endocrine secretion [1]. This leads to serious pathophysiological changes such as depression, anxiety, sleep disorders, hot flashes, night sweats, and menstrual disorders [2]. In addition, the risk of metabolic abnormalities in menopausal women increases significantly [3–6]. For example, a cross-sectional and longitudinal analysis has shown that plasma intermediate-density lipoprotein (IDL) and LDL levels are higher in menopausal women than in women of childbearing age [3]. Another prospective population study has shown that menopause and aging are independently correlated with body mass index (BMI) increase [4]. Insulin resistance in menopausal women is found to increase the incidence of type II diabetes (T2DM) [5], and significantly increase the risk of metabolic syndromes (MetS) [6]. Thus, the decline in estrogen level in menopause is closely related to metabolic abnormalities.

Oxidative stress is defined as an imbalance between oxidation and anti-oxidation in the body where the endogenous antioxidants are not able to counteract the oxidative dysregulation of lipids, protein, DNA and cellular structures [7]. Oxidative stress has been found to be closely related to metabolism in recent years [8, 9]. For instance, high concentration of hydrogen peroxide ($H_2O_2$) promotes insulin signaling, inducing insulin-like metabolism to increase glucose uptake by adipocytes and muscles and stimulating glucose transporter type-4 (GLUT-4) translocation and lipid synthesis in adipocytes [8]. In obese populations, the abnormal secretion of anti-inflammatory factors, such as resistin, visfatin, and adiponectin, and pro-inflammatory factors, such as tumor necrosis factor alpha (TNF-α), interleukin (IL)-1 and IL-6, in adipose tissues causes an increase in reactive oxygen species (ROS) in the body [9].

Nuclear factor E2-related factor 2 (NRF2) is an oxidative stress protein sensor that plays critical roles in oxidative defense, inflammatory reaction, and anti-apoptosis in cells [10]. Recent studies showed that NRF2 also affects energy metabolism and activates extracellular signal-regulated kinase (ERK) signaling to participate in insulin resistance in cardiomyocytes; activated NRF2 enhances insulin receptor sensitivity and increases glucose uptake [11]. In addition, a significant reduction of abdominal fat was found in *Nrf2*-knockout mice, and the mRNA expression of genes involved in fat synthesis in *Nrf2*-knockout mice was reduced, indicating that NRF2 plays a negative regulatory role in lipid metabolism [12]. However, another study showed markedly accelerated adipogenesis upon stimulation in *Nrf2* -/- embryonic fibroblast cells (MEFs), while *Keap1* -/- MEFs differentiated slowly compared to their congenic wild-type MEFs [13]. Therefore, the effects of *Nrf2* gene on energy metabolism in different models require further clarification.

Menopause changes the metabolic state due to changes in the endocrine environment. As energy metabolism is closely related to oxidative stress, *Nrf2* gene deletion may affect the metabolism of menopausal women through various signaling pathways. Ovariectomy is the most common method of establishing an animal model of menopause. Hence, this study established an *Nrf2*-knockout menopausal mouse model to evaluate the effect of *Nrf2* deficiency on glucose and lipid metabolism in menopausal females. The results of this study further our understanding of the molecular mechanisms and genetic basis of metabolic disorders in menopausal women.

## Methods

### 1. Experimental subjects and grouping

This animal study was approved by the Ethics Committee of the Third Affiliated Hospital of Guangzhou Medical University (No.2018[026]). Experiments were conducted according to

the National Institutes of Health guide for the care and use of Laboratory animals, and all efforts were made to minimize animal suffering. Specific-pathogen free wide-type C57BL/6J female mice were purchased from the Guangdong Medical Laboratory Animal Center (Guangzhou, China). *Nrf2*-deficient mice were purchased from the Jackson Laboratory (Bar Harbor, ME, USA) on an 129X B6 F1 background and were then backcrossed onto a C57BL6/J strain for more than 10 generations [14]. All female mice were housed in the specific pathogen-free animal facility in the Laboratory Animal Center of South China Agriculture University (Guangzhou, China), and maintained on a controlled light cycle schedule of 12:12 h (light/dark) at 24°C with chow diet ad libitum. Genotypes of all animals were determined by analysis of DNA extracted from mouse tails.

The female mice were divided into four groups (7–8 animals for each group): wild-type control (WT-CON), wild-type ovariectomy (WT-OVX), *Nrf2−/−* control (KO-CON) and *Nrf2−/−* ovariectomy (KO-OVX). Ovariectomy was performed in the WT-OVX and KO-OVX groups once the mice reached 15–16 weeks old, while a simulated operation was performed in the WT-CON and KO-CON groups. For surgery, mice were anaesthetized with an IP injection of 100 mg/kg ketamine, and postoperative recovery of all animals was monitored. Vaginal smears were performed in all groups of female mice for seven consecutive days at four weeks postoperatively, followed by staining the smear samples with methylene blue to observe changes in the estrous cycle in mice [15].

## 2. Experimental sample collection and preparation of bodyweight curve

All experimental mice were weighed before surgery and in the postoperative 1, 3, 5, 7, 9 weeks after overnight fasting. About 100 ul of blood samples were collected through retro-orbital (eyelid blood collection) before surgery and at the end of each time point postoperatively. When sacrificed at postoperative 9 week, all animals were euthanized using $CO_2$ and 500 ul of blood was collected by eyeball dissection, followed by dissecting the brain on ice and storage of the tissues at −80°C for later use. The collected peripheral blood samples were let stand at room temperature for an hour, followed by centrifuging at 775g for 15 min to collect and storage of the serum at −80°C for later use.

## 3. Detection of estrogen levels

Serum hormone estradiol (E2) form preoperative and postoperative week 5 was detected using an enzyme-linked bio-estradiol (E2) ELISA kit (Shanghai Enzyme-linked Biotechnology Co., Ltd., Shanghai, China) by adding 50 μl of individual standards and samples into each standard and sample well, respectively, and not adding any solution in the blank controls. After adding 100 μl horseradish peroxidase (HRP)-labeled E2 antibodies into each well to incubate at 37°C for 60 min, the ELISA plate was washed and 50 μl of coloring solutions A and B were independently added into each well to incubate at 37°C for 10 min. After adding 50 μl stopping solution to terminate the reaction, the absorbance of each well was measured with an ELx808 microplate reader (Biotek, USA) to calculate the sample concentrations.

## 4. Detection of glucose and lipid metabolism indices

Serum low-density lipoprotein (LDL), high-density lipoprotein (HDL), triglycerides (TG), total cholesterol (T-CHO) and glucose (GLU) were measured using the corresponding detection kits (Jiancheng Bioengineering Institute, Nanjing, China) preoperatively and at postoperative weeks 1, 3, 5, 7, and 9, strictly in accordance with the manufacturer's instructions. The detection range, sensitivity, and the intra and inter assay coefficient of variation of all biochemical and ELISA kits were provided in S1 Table.

## 5. Oxidative stress marker, malondialdehyde (MDA)

Serum MDA concentration at postoperative week 9 was measured using an MDA kit (Cloud-Clone Corp., Wuhan, China) by adding 50 μl standards or serum samples into standard or blank control wells, respectively. Then, 50 μl sample diluting solution was added to the blank control well. Subsequently, 50 μl of biotin-labeled antigen was added to each well to incubate at room temperature for 60 min, followed by washing each well and adding 100 μl HRP-labeled avidin to incubate for 30 min. After washing the ELISA plate, 90 μl color developing solution was added to each well to incubate for 10 min, followed by adding 50 μl stopping solution per well to stop the reaction and reading the absorbance using an ELx808 microplate reader (Biotek, USA) to calculate the individual sample concentration.

## 6. Detection of neurotransmitters

Brain tissues stored at −80˚C were weighed and cell lysis buffer (0.1% SDS + 0.05% Triton-X) was added in a weight ratio of 1 g to 9 ml buffer for homogenization. The brain homogenates were then centrifuged at 3,000 rpm for 20 min to collect the supernatants for the detection of mouse dopamine (DA) and serotonin (5-hydroxytryptamine, 5-HT), using the DA and 5-HT ELISA kits (Shanghai Enzyme-linked Biotechnology Co., Ltd, Shanghai, China) strictly in accordance with the manufacturer's instructions.

## 7. Statistical analysis

All statistical analyses were performed using Graphpad Prism 5.0 (La Jolla, CA, USA). Data are presented as mean ± standard error (SEM). One-way ANOVA with Bonferroni's Multiple comparison test or Kruskal-Wallis test with Dunn's Multiple comparison test was used for ordinary or non-parametric data, as specified in the figure legends. Statistical significance was defined as $P < 0.05$.

## Results

### 1. Changes in estrogen levels and the estrous cycle in different groups of mice

This study compared the preoperative and postoperative estrogen levels and calculated the changed estrogen rate of each mouse at postoperative week 5. The results showed that the change rate of estrogen levels of the WT-OVX and KO-OVX groups were significantly lower than that of the WT-CON and KO-CON groups (WT-CON 60.12% ± 124.9%, WT-OVX -21.02% ± 19.32%, KO-CON 34.03% ± 47.50%, and KO-OVX -27.02% ± 14.19, $P < 0.05$). No significant changes in different genotypes of the ovariectomy and control groups were found (Fig 1a). In addition, postoperative methylene blue staining of the continuous vaginal smears of the WT-OVX and KO-OVX groups demonstrated a change in the estrous condition, as a large number of white blood cells or a small number of keratinized epithelial cells were observed (Fig 1b). However, normal estrous changes were observed in the mice of the WT-CON and KO-CON groups, and a large number of keratinized epithelial cells were observed under light microscopy (Fig 1b).

### 2. Comparison of MDA levels in different groups of mice

MDA is one of the products of oxidative stress, reflecting the peroxidation level of the body. This study observed the changes in oxidative stress levels in different groups by detecting the serum MDA level of mice. The results showed that, compared to the WT-CON group, the

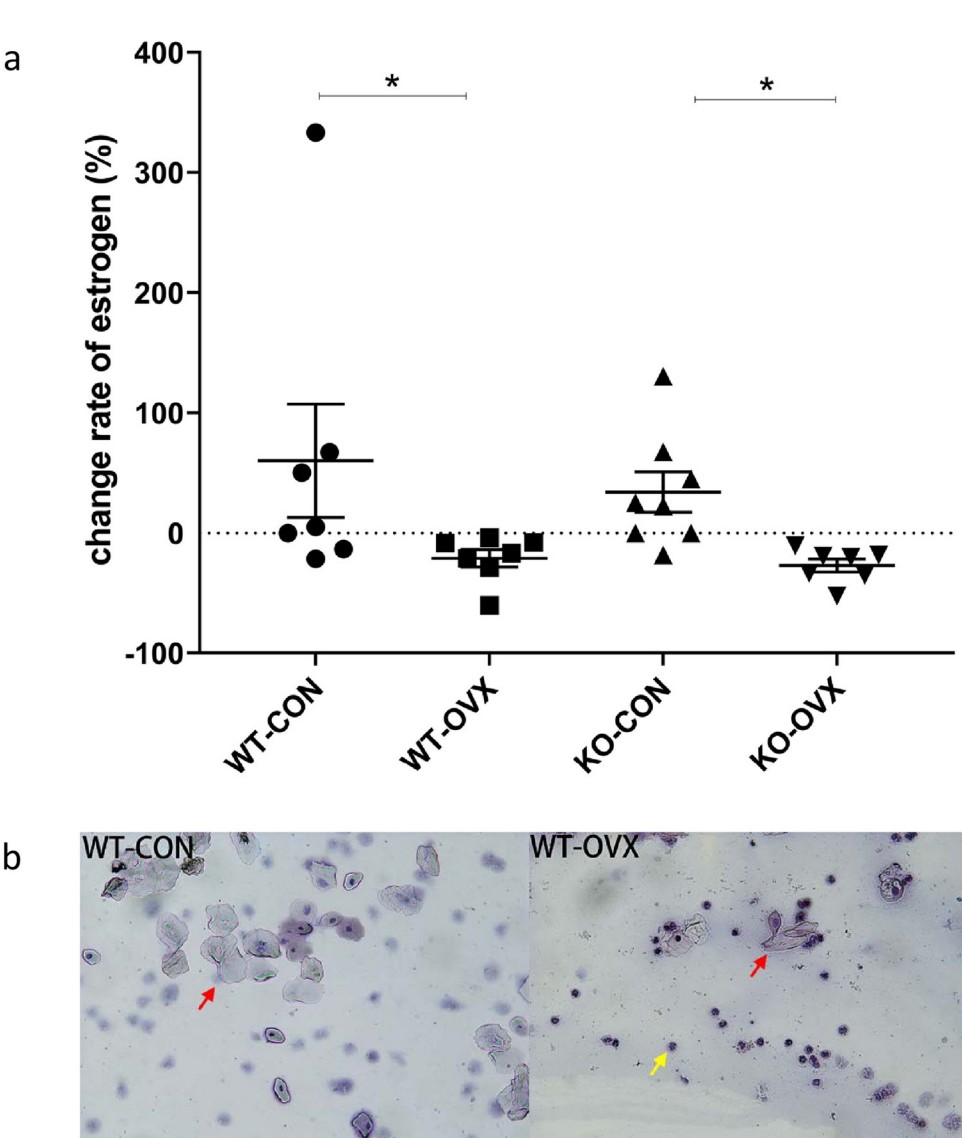

**Fig 1. Changes of estrogen levels and vaginal smears in different groups of mice.** WT-CON: wild-type control,
WT-OVX: wild-type ovariectomy, KO-CON: *Nrf2*−/− control, KO-OVX: *Nrf2*−/− ovariectomy. (a) The change rate of
estrogen levels before ovariectomy and at postoperative week 5. Results are shown as mean ± SEM. * $P < 0.05$ analyzed
by Kruskal-Wallis test with Dunn's Multiple comparison test. (b) Results of vaginal smears in different groups of mice
at postoperative week 4–5. Vaginal smears in the left figures represent the morphologies of estrus, with a large number
of keratinized epithelial cells observed under light microscopy. Vaginal smears in the right figures represent the
morphologies of diestrus, with a large number of white blood cells and a small number of keratinized epithelial cells.
The red arrows represent keratinized epithelial cells and the yellow arrows represent white blood cells.

serum MDA levels increased in the WT-OVX and KO-CON groups, but decreased in the KO-OVX group, though no statistical significance was reached yet (Fig 2).

### 3. Body weight changes in different groups of mice

Body weight curves prepared by weighing each group of mice weekly showed that weight gain occurred in the WT-OVX and KO-OVX groups relative to the WT-CON and KO-CON groups (Fig 3b). No significant difference of the weight gain rate was found between the WT-CON and KO-CON groups, but OVX groups had increased weight compared to the counterpart CON groups (Fig 3b and 3a, $P < 0.05$ and $P < 0.01$). By calculating the rate of weight gain from the preoperative time to postoperative week 9 in different groups, we found that the weight gain rates of the KO-OVX was significantly higher than that in the corresponding KO-CON and WT-CON groups after postoperative week 4 (Fig 3b), and the weight gain rate of KO-OVX was the greatest and was significantly different from the WT-CON and KO-CON group (WT-CON 6.02% ± 2.95%, WT-OVX 18.55% ± 7.40%, and KO-CON 5.25% ± 5.08%, KO-OVX 27.35% ± 10.04%).

### 4. Time-varying maps of metabolic indicators in different groups of animals

The changes in the serum LDL, HDL, TG, T-CHO, and GLU concentrations were measured preoperatively and at postoperative weeks 1, 3, 5, 7, and 9, and compared in different groups of mice (Fig 4a–4e).

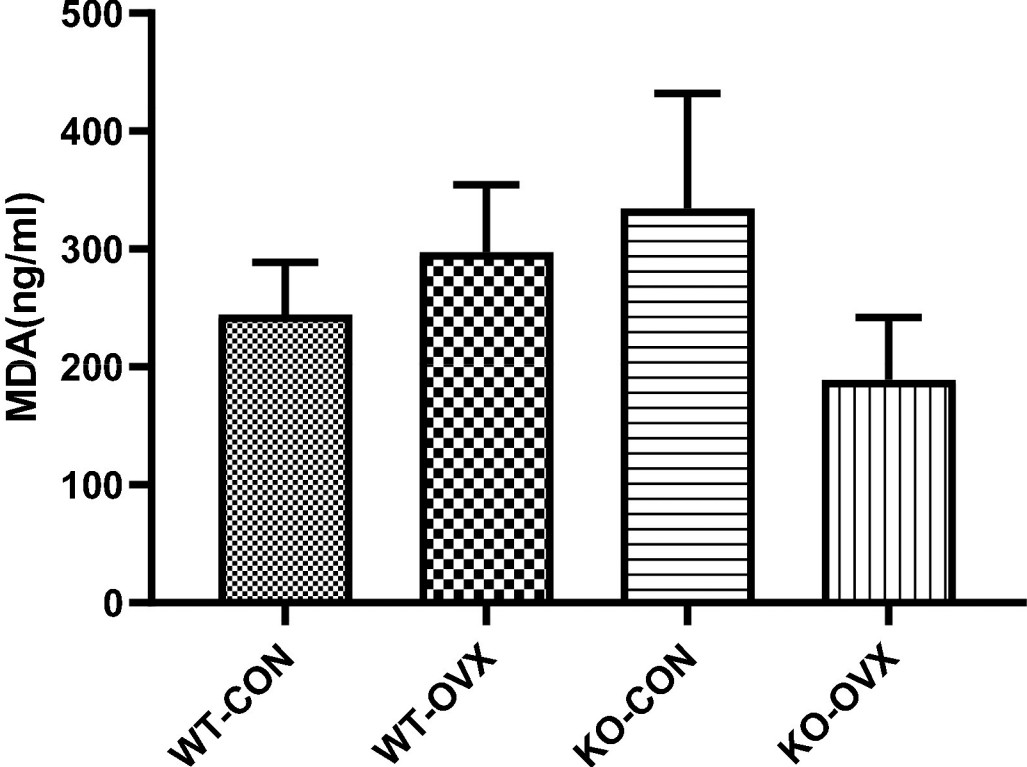

**Fig 2. Comparison of malondialdehyde (MDA) levels in different groups of mice.** WT-CON: wild-type control, WT-OVX: wild-type ovariectomy, KO-CON: *Nrf2*−/− control, KO-OVX: *Nrf2*−/− ovariectomy. Serum MDA concentration at postoperative week 9 was measured in different groups. Results are shown as mean ± SEM.

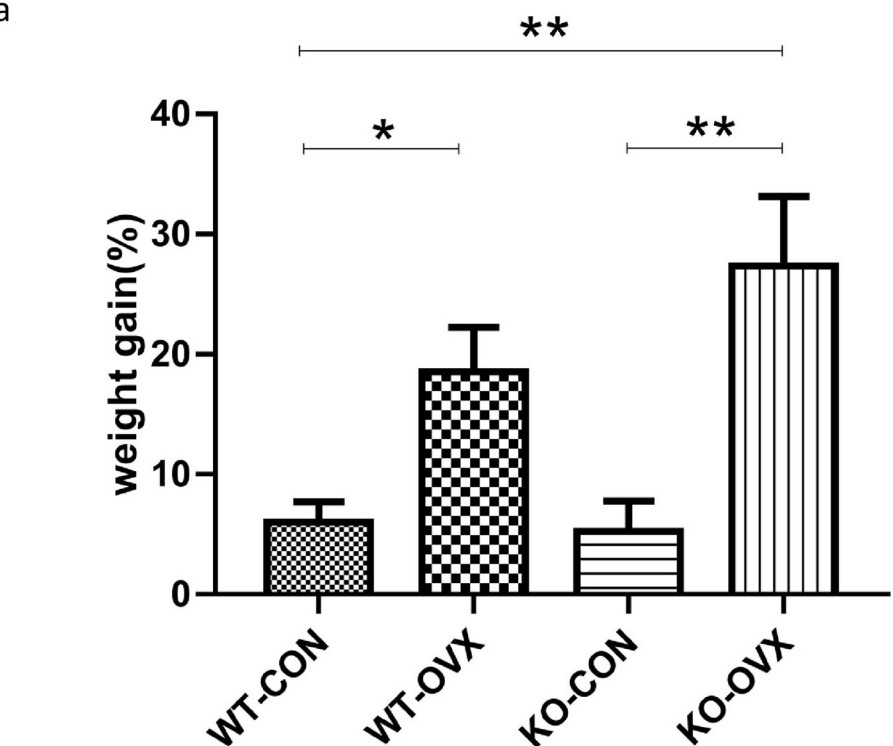

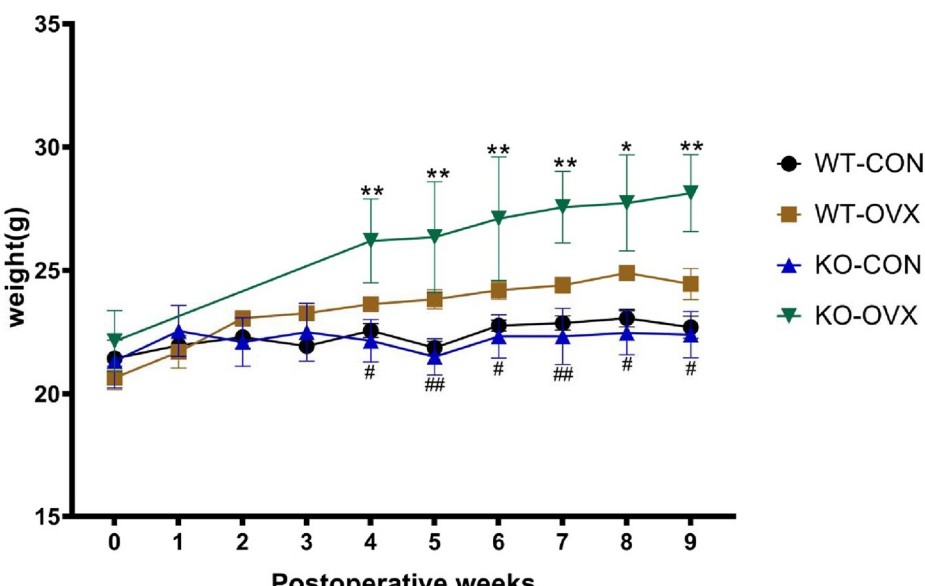

**Fig 3. Changes in bodyweight in different groups of mice.** WT-CON: wild-type control, WT-OVX: wild-type ovariectomy, KO-CON: *Nrf2−/−* control, KO-OVX: *Nrf2−/−* ovariectomy. (a) The weight gain in different groups of mice preoperatively and at postoperative week 9. $^{*}P < 0.05$; $^{**}P < 0.01$. (b). Changes in bodyweight curve of different groups of mice during 9 weeks after ovariectomy. $^{*}P < 0.05$; $^{**}P < 0.01$ (KO-OVX vs KO-CON); $^{\#}P < 0.05$; $^{\#\#}P < 0.01$ (KO-OVX vs WT-CON). Results are shown as mean ± SEM. Data were analyzed by One-way ANOVA with Bonferroni's Multiple comparison test.

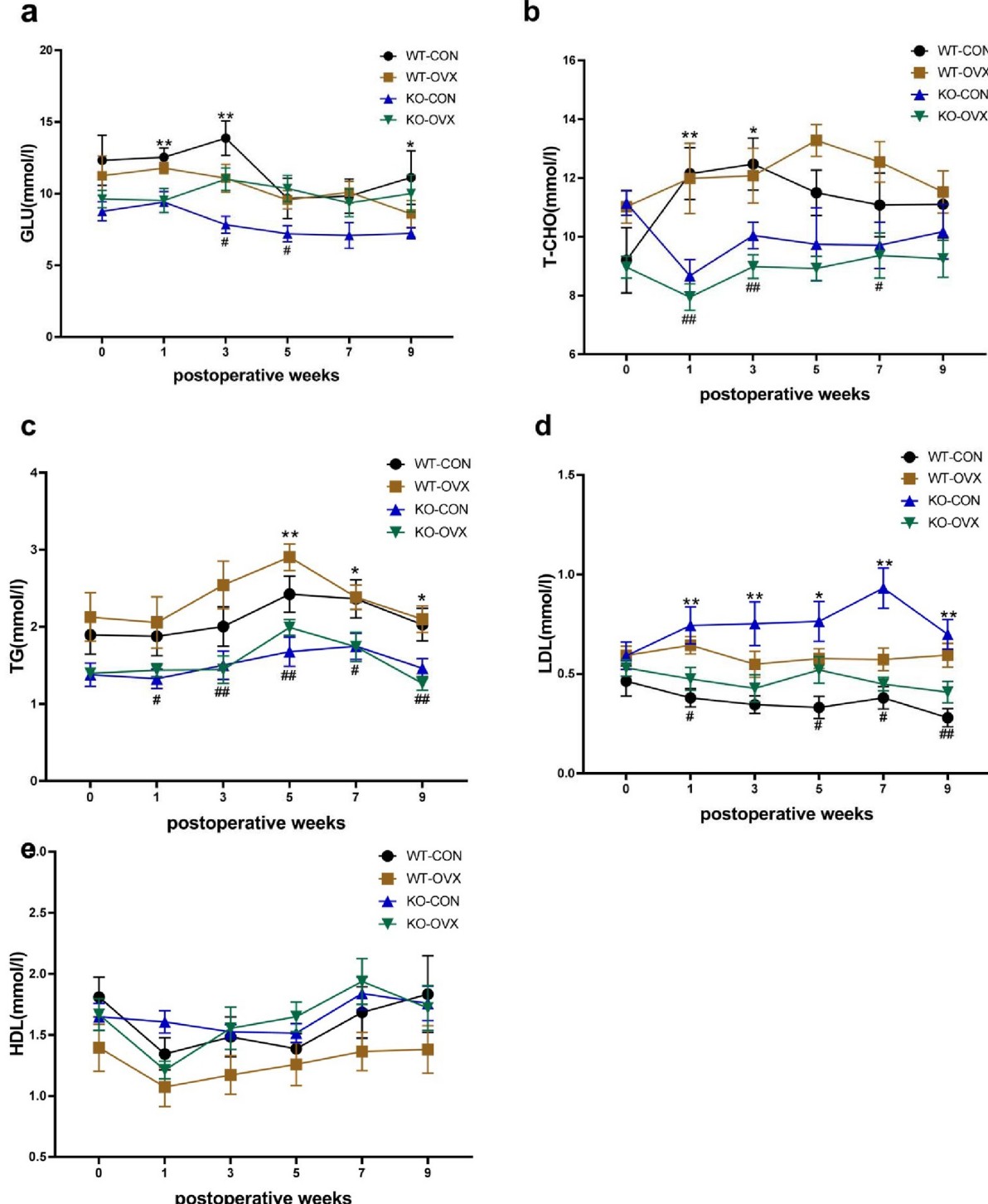

**Fig 4. Changes in different metabolic indicators in different groups of mice over time.** WT-CON: wild-type control, WT-OVX: wild-type ovariectomy, KO-CON: *Nrf2–/–* control, KO-OVX: *Nrf2–/–* ovariectomy. Curves showing the changes in serum metabolite levels preoperatively (week 0) and at postoperative week 1, 3, 5, 7, and 9 in different groups of mice. (a) Change curve of glucose levels. $^{*}P < 0.05$; $^{**}P < 0.01$ (KO-CON vs WT-CON); $^{\#}P < 0.05$ (KO-OVX vs KO-CON). (b) Change curve of T-CHO levels. $^{*}P < 0.05$; $^{**}P < 0.01$ (KO-CON vs WT-CON); $^{\#}P < 0.05$; $^{\#\#}P < 0.01$ (KO-OVX vs WT-OVX). (c) Change curve of TG levels. $^{*}P < 0.05$; $^{**}P < 0.01$ (KO-CON vs WT-CON); $^{\#}P < 0.05$; $^{\#\#}P < 0.01$ (KO-OVX vs WT-OVX). (d) Change curve of LDL levels. $^{*}P < 0.05$; $^{**}P < 0.01$ (KO-OVX vs KO-CON); $^{\#}P < 0.05$; $^{\#\#}P < 0.01$ (WT-OVX vs WT-CON). (e) Change curve of HDL levels. Results are shown as mean ± SEM. Data were analyzed by One-way ANOVA with Bonferroni's Multiple comparison test.

Our results showed that the serum GLU of the KO-CON group was significantly lower than that of the WT-CON group at postoperative weeks 1, 3, and 9 (Fig 4a, $P < 0.05$), and also had low tendency at postoperative weeks 5, and 7. However, the serum GLU of the KO-OVX group was significantly higher than that of the KO-CON group at postoperative week 3 and 5 ($P < 0.05$), and was marginally higher at postoperative week 7 and 9, while there was little change between WT-OVX and WT-CON groups (Fig 4a).

The serum T-CHO levels of the KO-CON group was significantly lower than that of the WT-CON group at postoperative weeks 1, 3, and 5 (Fig 4b, $P < 0.05$), and that of the KO-OVX groups were significantly lower than that of the WT-OVX groups after ovariectomy (Fig 4b, all $P < 0.05$). However, there was little change of serum T-CHO levels between OVX groups and counterpart CON groups in WT and KO mice (Fig 4b).

In addition, the KO-CON group also had significantly lower levels of serum TG compared to the WT-CON group at postoperative weeks (Fig 4c, $P < 0.05$), and that of the KO-OVX group were significantly lower than that of the WT-OVX group after ovariectomy (Fig 4c, all $P < 0.05$), and. However, there was little change between CON groups and counterpart OVX groups in WT and KO mice respectively (Fig 4c).

Comparison between the KO and WT groups showed that the serum LDL levels of the KO-CON group was significantly higher than that of the WT-CON group during postoperative period (Fig 4d, all $P < 0.05$). Comparison the preoperative groups and counterpart postoperative groups, the LDL levels in the WT-OVX significantly increased (Fig 4d, all $P < 0.05$), but the LDL levels in the KO-OVX group significantly decreased after ovariectomy (Fig 4d, all $P < 0.05$). However, the serum HDL levels of has no statistically significant difference among all groups (Fig 4e).

## 5. Changes in neurotransmitters in different groups

To evaluate the effect of menopause on the mental status, this study examined the levels of 5-HT and DA in brain tissues. The inhibitory neurotransmitter, 5-HT, of the KO-OVX group decreased after ovariectomy compared to the KO-CON group, which was significantly lower than that of the WT-CON and WT-OVX groups (Fig 5a, $P < 0.05$). The test value (ng/ml) showing as 37.88 ± 13.06, 37.18 ± 14.24, 31.54 ± 8.39, and 25.07 ± 3.25 in the WT-CON, WT-OVX, KO-CON, and KO-OVX groups, respectively. Another excitatory neurotransmitter, DA, was found no significant difference in the brain tissues in different groups (Fig 5b).

## Discussion

This study used ovariectomy to establish a menopausal mouse model and showed that the KO menopausal mouse model had significantly more weight gain, and significantly different glucose and LDL metabolism than the WT menopausal mice. In addition, the 5-HT levels in the brains of the KO menopausal mouse model were significantly lower than that of the WT menopausal mice. Therefore, an *Nrf2* deletion is a genetic factor that causes susceptibility to menopausal female obesity and its induction mechanism is possibly different from a simple decline in estrogen level.

After ovariectomy surgery, both KO mice and WT mice had a significant decrease in estrogen levels (Fig 1). Numerous studies have shown that estrogen can effectively regulate weight gain [16–18]. In a longitudinal study, increased waist circumference, BMI, and fat mass and reduced skeletal muscle mass were found in menopausal women, which are associated with changes in menopausal hormone levels [19]. A study has shown a similar conclusion that postmenopausal women are five times more likely to have central obesity than premenopausal women [20]. In this study, we found that the body weight gain of the menopausal mouse

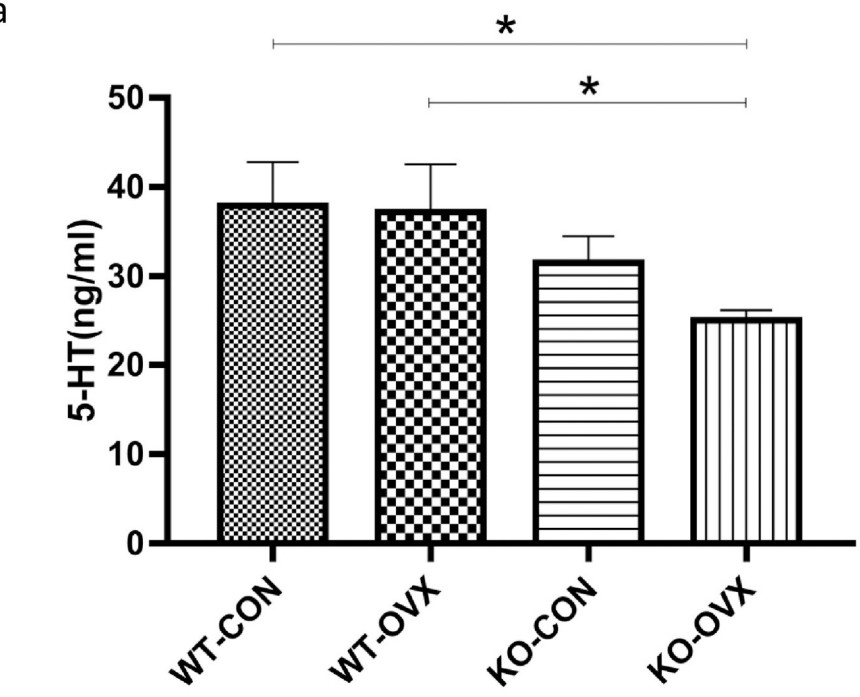

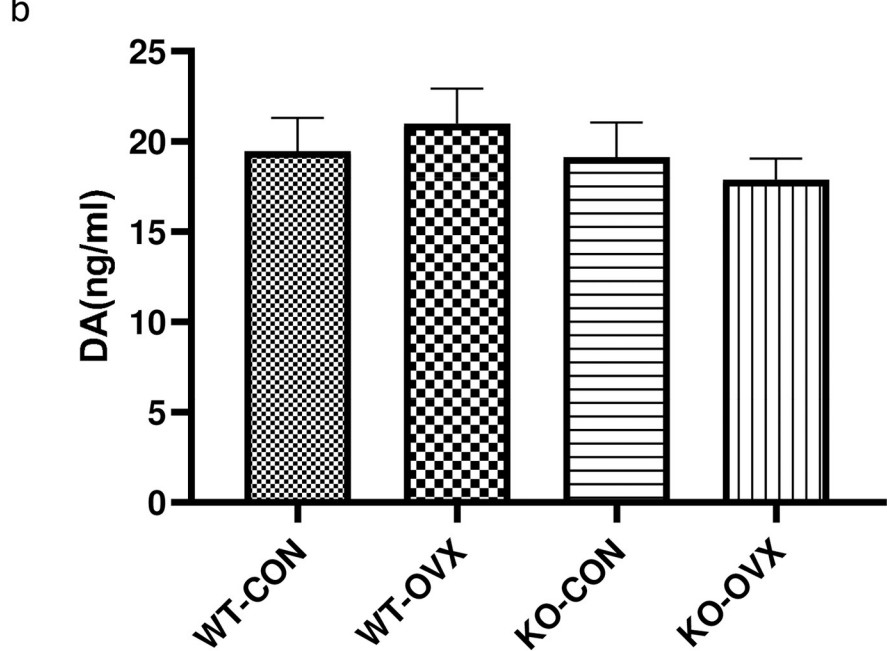

**Fig 5. Concentrations of neurotransmitters in the brain tissues of different groups of mice.** WT-CON: wild-type control, WT-OVX: wild-type ovariectomy, KO-CON: *Nrf2–/–* control, KO-OVX: *Nrf2–/–* ovariectomy. (a) Concentrations of 5-hydroxytryptamine (5-HT) in the brain tissues of different groups of mice at postoperative week 9. (b) Dopamine (DA) concentrations in the brain tissues of different groups of mice at postoperative week 9. Results are shown as mean ± SEM. *$P < 0.05$ analyzed by One-way ANOVA with Bonferroni's Multiple comparison test.

models showed a significant increase compared with the control groups, with the body weight increasing by 18%–27% at postoperative week 9, which proves that a decline in estrogen level has a negative effect on bodyweight after menopause (Fig 3).

Studies of molecular mechanisms of menopausal weight gain have shown menopausal weight gain is closely related to the changes in lipid metabolism. A population survey has confirmed that T-CHO, LDL, and TG peak in menopausal women, which is associated with decreased E2 and increased follicle stimulating hormones [21]. In addition, the changes in T-CHO and LDL are significantly related to the changes in bodyweight [21]. Our data showed that the LDL level of the WT ovariectomized mice was significantly higher than that of the WT control mice, but that of T-CHO and TG did not reach statistical significance (Fig 4b, 4c and 4d). In addition, in menopausal animal model, it was reported that ovariectomy may have an effect on glucose metabolism and insulin resistance [22]. In our study, no significant change in blood glucose levels was found before and after ovariectomy in WT mice, however, in KO mice, ovariectomy led to significantly elevated glucose levels, which may partially account for their rapid weight gains.

Comparison of the metabolic data between KO control group and WT control group demonstrated that the lipid metabolism was significantly different (Fig 4). The T-CHO and TG levels of the KO mice were significantly lower, while the LDL level of the KO mice was significantly higher than that of the WT mice, suggesting *Nrf2* plays important roles in the regulation of lipid metabolism (Fig 4b, 4c and 4d). Previous studies have shown that a *Nrf2* deletion improves insulin resistance, inhibits lipid synthesis and regulates the expression of regulators of lipid synthesis [23]. In addition, these *Nrf2* KO mice had reduced fat mass, smaller adipocytes, inhibited lipogenesis in adipocytes [24, 25], and decreased blood glucose and cholesterol in serum, liver and adipose tissues [12, 26]. Therefore, *Nrf2* deletion effectively inhibits lipid synthesis in mice model before overiecomy (Fig 4b, 4c and 4d).

However, once the KO mice were ovariectomized, the T-CHO, TG and HDL levels had no significant changes compared with the KO control mice (Fig 4b, 4c and 4d). Notably, LDL levels in the KO mice were dramatically reduced after ovariectomy, when the estrogen level was reduced; while the LDL levels of the WT mice were sharply increased after the ovariectomy (Fig 4d). Although a dual role of NRF2 during metabolic dysregulation was reported as increasing lipid accumulation in liver and white adipose tissue but preventing lipid accumulation in obese mice [27], the exact LDL levels in *Nrf2*-knockout mice had no report so far. In this study, unchanged levels of the T-CHO, TG and HDL but elevated LDL was observed in KO mice after ovariectomy suggested that a decline in estrogen levels did not seem to completely reverse the inhibition of lipid synthesis caused by *Nrf2* deletion. For further elucidation, the adiposity of these animals should be evaluated to explain whether a greater weight gain in KO ovariectomized mice also mean fat accretion.

Previous studies have reported that 5-HT levels in cerebral spinal fluid reduced in a menopausal mouse model [28], and serum 5-HT concentrations were negatively associated with age, weight, BMI, fat mass in postmenopausal women [29]. In this study, we did not find significant alteration of 5-HT in WT mice after ovariectomy in the brain tissue homogenate (Fig 5a). However, KO menopausal mice had a significant reduction of 5-HT compared to WT groups (Fig 5a). As an inhibitory neurotransmitter, 5-HT is involved in energy metabolism by not only limiting food intake by suppressing appetite, but also participating in energy metabolism [30], and drugs that enhance 5-HT delivery have been extensively used to clinically treat obesity [31]. Therefore, the increased bodyweight of mice in the KO-OVX observed in this study might be associated with the significant reduction of brain 5-HT level. As the combination of *Nrf2* deletion and a decline in estrogen level leads to a more pronounced decrease in 5-HT levels, thereby increasing the susceptibility to obesity. Therefore, whether *Nrf2* deletion in ovariectomized female mice can alter their food intake and cause hyperphagia should to be further confirmed.

Although global knock-out of *Nrf2* in mice protects against weight gain and obesity, on different background, the exact effects of *Nrf2* are still controversial [12, 32, 33]. For instance, in diet-induced obesity mice model, adipose-specific *Nrf2* KO mice showed reduced blood glucose, reduced number but increased size of adipocytes [34]. In the Lepob/ob mice (OB) model, ablation of *Nrf2* led to reduced white adipose tissue mass, but resulted in an even more severe metabolic syndrome with aggravated insulin resistance, hyperglycemia, and hypertriglyceridemia [23, 27]. Based on current data, it was supposed that the superposition of *Nrf2* deficiency and a decline in estrogen level promoted weight gain *in vivo* (Fig 3), and there might be complex interactions between these two molecules on metabolism regulation.

Recently, increasing evidence has demonstrated that *Nrf2* signaling is antagonistic to estrogen signaling in hepatic fat metabolism [35, 36]. For instance, a high-fat diet and simultaneous administration of exogenous estrogen reduced the hepatic TG levels by 49% and 90% in WT and KO nonalcoholic fatty liver disease mice, respectively, which suggests that *Nrf2* deletion significantly amplifies the inhibitory effect of estrogen on adipogenesis [35]. At the same time, administration of 17β-E2 in ovariectomized mice significantly reduces ROS production and upregulates *Nrf2* mRNA and protein expression, which induces the expression of phase II antioxidant enzymes [36]. Therefore, besides *Nrf2* deficiency and estrogen are both inhibitors for weight gain, they might also have interactive regulation in the energy metabolism. Based on current data, we did not know the detailed mechanisms for observed interactions yet. In addition, as *Nrf2* also coordinates the basal and stress-inducible activation of a vast array of cytoprotective genes, what are their effects on the menopausal metabolism is not uncertain and awaits further research.

## Conclusions

This study showed that compared with the corresponding WT mice, *Nrf2* KO ovariectomized mice had a greater bodyweight gain. Different from their WT counterparts, a significant increase in blood glucose level, unchanged T-CHO, TG, and HDL levels, but a significant reduction in serum LDL was found in the *Nrf* KO mice after ovariectomy. In addition, the level of the neurotransmitter 5-HT in the brain was significantly reduced in the *Nrf2* KO mice after ovariectomy. Thus, the combination of *Nrf2* deletion and a decline in estrogen level induced a significant increase in bodyweight. The mechanism may be different from weight gain caused by a simple decline in estrogen level but may be associated with the altered glucose and LDL metabolism regulation and decreased 5-HT levels.

## Supporting information

**S1 Data.**
(XLSX)

**S1 Table. Quality control parameters of biochemical and ELISA assays.**
(DOCX)

## Acknowledgments

We thank LetPub (www.letpub.com) for its linguistic assistance during the preparation of this manuscript.

## Author Contributions

**Data curation:** Cong Shen, Yeling Liu.

**Formal analysis:** Shengjie He, Junquan Sun.

**Funding acquisition:** Bolan Yu.

**Supervision:** Jun Huang.

**Writing – original draft:** Xunwei Wu.

**Writing – review & editing:** Bolan Yu.

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
