## [Decision Letter · Decision Letter 0]

4 Nov 2019

PONE-D-19-25429

NRF2 deficiency increases obesity susceptibility in a mouse menopausal model

PLOS ONE

Dear Mrs. Yu,

Thank you for submitting your manuscript to PLOS ONE. After careful consideration, we feel that it has merit but does not fully meet PLOS ONE’s publication criteria as it currently stands. Therefore, we invite you to submit a revised version of the manuscript that addresses the points raised during the review process.

Two different reviewers with expertise in the study subject agreed that a major review is needed. Please pay special atrention to statistical analysis, which must be changed as well as other comments on methods and results/ discussion. 

We would appreciate receiving your revised manuscript by Dec 19 2019 11:59PM. To enhance the reproducibility of your results, we recommend that if applicable you deposit your laboratory protocols in protocols.io, where a protocol can be assigned its own identifier (DOI) such that it can be cited independently in the future. For instructions see: http://journals.plos.org/plosone/s/submission-guidelines#loc-laboratory-protocols

We look forward to receiving your revised manuscript.

Kind regards,

Vanessa Souza-Mello, Ph.D.

Academic Editor

PLOS ONE

Journal Requirements:

2. To comply with PLOS ONE submission requirements, in your Methods section, please provide additional information regarding the experiments involving animals and ensure you have included details on 1) methods of anesthesia and/or analgesia, and 2) efforts to alleviate suffering.

3. PLOS journals require authors to make all data underlying the findings described in their manuscript fully available without restriction at the time of publication. As such, please supply the minimal data set as a new supplementary figure/table, or deposit it in a public database. (By minimal data set, we mean the values behind the graphs and charts shown in your manuscript.) For more information, please see our data availability policy at https://journals.plos.org/plosone/s/data-availability.

Reviewers' comments:

Reviewer's Responses to Questions

**Comments to the Author**

1. Is the manuscript technically sound, and do the data support the conclusions?

Reviewer #1: Yes

Reviewer #2: Yes

2. Has the statistical analysis been performed appropriately and rigorously? 

Reviewer #1: Yes

Reviewer #2: No

3. Have the authors made all data underlying the findings in their manuscript fully available?

Reviewer #1: Yes

Reviewer #2: Yes

4. Is the manuscript presented in an intelligible fashion and written in standard English?

Reviewer #1: Yes

Reviewer #2: Yes

5. Review Comments to the Author

Reviewer #1: The article entitled “Nrf2 deficiency increases obesity susceptibility in a mouse menopausal model” explains the importance of Nrf2 on metabolic abnormalities in menopausal women using overiectomized mouse menopausal model. The authors show the importance of Nrf2 gene in onset of obesity, which is already known with several published articles. Obesity itself has shown to reduce Nrf2 expression, as well as Nrf2 deletion alters estrus cycle, which is validated in nrf2-/- mice. The mechanistic aspects of Nrf2 deletion on menopause is lacking. The manuscript needs some revisions,

1. The authors need to add the references for estradiol levels, estrus cycle in Nrf2 deletion.

2. It is known that overiectomy is done in female mice, but the authors need to mention the mouse gender used in this study?

3. In the methodology the authors don’t mention time points study but in the results section they have results at different time points (Fig. 4). This needs to be explained in the methodology.

4. The collection of blood through retro-orbital (eyelid blood collection) which is mentioned in the methods is for end point or terminal experiments. Did the authors collect blood through this method at the end of each time point? If not how much of the serum did they use for the detection of glucose and lipid metabolism indices at each time point?

5. Standard deviation in most of the results is more than the mean values? (Fig. 1, Fig. 3 & Fig. 4)

6. Fig. 3b, change in body weights significance between the groups needs to be shown?

7. In the discussion, there is a sentence which says “ numerous studies have shown that estrogen……” but only one reference quoted?

Reviewer #2: In the present study, the authors investigated the role of the Nuclear factor E2-related factor 2 (NRF2), an important oxidative stress sensor that plays regulatory role in energy metabolism, on metabolic parameters of ovariectomized menopausal model.

This study is of interest as it addresses an important biological and medical issue. On the whole, the manuscript is sound and well written but I however have some minor concerns that should be addressed by the authors before publication.

Abstract: please include p-value after differences affirmations through the abstract.

Materials & methods

The authors took in account the the National Institutes of Health guide for the care and use of Laboratory animals? Please, provide this information

Statistical analysis: in the present study the authors compared four groups, T-test seems to not be the best way to perform this analysis. ANOVA must be applied.

Please, review all numbered sections through the manuscript.

Results.

Please, provide food intake/energy intake of the animals. Is it hyperphagia presented in ovariectomized mice?

4. Time-varying maps of metabolic indicators in different groups of animals: what the author’s means with “various metabolic substances”? Unclear… please rephrase

Materials & method.

Centrifugation speed should be expressed as g instead of rpm as rpm depends on the diameter of the rotor. Of the centrifuge. Table is available on the web to do this conversion using the speed in rpm and the diameter of the rotor.

Materials & methods: Please specify the sensitivity of the assay (i.e. the LOD) and the intra assay coefficient of variation of all ELISA assay.

Were the main white adipose tissue fat pads dissected out to evaluate the adiposity of the animals. Does a greater weight gain in the groups also mean fat accretion?

If the adipose tissue accretion is reduced in some groups (here we need food intake to discuss) there is an “energetic paradox” according to the first law of thermodynamics… (Energy intake = Energy expenditure + Energy stored)

Nuclear factor E2-related factor 2 (Nrf2) is a transcription factor that coordinates the basal and stress-inducible activation of a vast array of cytoprotective genes. Understanding the regulation of Nrf2 activity and downstream pathways has major implications for human health, include in menopause woman.

In the presented work, the authors showed the impact of NRF2 depletion on oxidative stress but not in antioxidant response. How this could limit the conclusion of the work?

Also in discussion section, the evidence of NRF2 and energy metabolism should be clearer.

6. PLOS authors have the option to publish the peer review history of their article (what does this mean?). If published, this will include your full peer review and any attached files.

Reviewer #1: No

Reviewer #2: No

---

## [Author Response · Author response to Decision Letter 0]

19 Dec 2019

Dear editor,

Thanks for timely review on our manuscript entitled "NRF2 deficiency increases obesity susceptibility in a mouse menopausal model". We have submitted the revised manuscript and marked copy with track changes in red. Below is the point-to-point response to the comments of the reviewers and editorial requests. Thank you very much for your kindly helps and looking forward to hear from you soon.

Best regards,

Bolan Yu

Point to point response

Editorial requests:

1.Two different reviewers with expertise in the study subject agreed that a major review is needed. Please pay special attention to statistical analysis, which must be changed as well as other comments on methods and results/ discussion.

We have revised the entire manuscript according to the reviewers’ comments, and statistical analysis was performed using One-way ANOVA.

Review Comments to the Author

Reviewer #1: The article entitled “Nrf2 deficiency increases obesity susceptibility in a mouse menopausal model” explains the importance of Nrf2 on metabolic abnormalities in menopausal women using overiectomized mouse menopausal model. The authors show the importance of Nrf2 gene in onset of obesity, which is already known with several published articles. Obesity itself has shown to reduce Nrf2 expression, as well as Nrf2 deletion alters estrus cycle, which is validated in nrf2-/- mice. The mechanistic aspects of Nrf2 deletion on menopause is lacking. The manuscript needs some revisions,

1. The authors need to add the references for estradiol levels, estrus cycle in Nrf2 deletion.

Based on our knowledge, we haven't found the reference for estradiol and estrus cycle in Nrf2 KO mice yet, nor did evidence suggest that Nrf2 deletion may cause altered estrus cycle. 

2. It is known that overiectomy is done in female mice, but the authors need to mention the mouse gender used in this study?

Thanks for the suggestion, we have added the gender for the mouse.

3. In the methodology the authors don’t mention time points study but in the results section they have results at different time points (Fig. 4). This needs to be explained in the methodology.

We have added the time points in the methods section (eg. Experimental sample collection and preparation of bodyweight curve).

4. The collection of blood through retro-orbital (eyelid blood collection) which is mentioned in the methods is for end point or terminal experiments. Did the authors collect blood through this method at the end of each time point? If not how much of the serum did they use for the detection of glucose and lipid metabolism indices at each time point?

Yes, the blood samples were collected at the end of each time point. About 100 ul of blood (which yields about 50 ul serum) were collected through retro-orbital. When sacrificed at postoperative 9 week, all animals were euthanized using CO2 and 500 ul of blood (which yields about 250 ul serum) was collected by eyeball dissection. For the detection of glucose and lipid metabolism, 2-3 ul serum was used for each experiment.

5. Standard deviation in most of the results is more than the mean values? (Fig. 1, Fig. 3 & Fig. 4)

Yes, according to raw data, these measures vary in different subjects and thus standard deviation may be more than the mean values. In the revised manuscript, we represented results ± SEM in all figures.

6. Fig. 3b, change in body weights significance between the groups needs to be shown?

We have added the * and # to showing the significance.

7. In the discussion, there is a sentence which says “ numerous studies have shown that estrogen……” but only one reference quoted?

We have added more reference.

Reviewer #2: 

In the present study, the authors investigated the role of the Nuclear factor E2-related factor 2 (NRF2), an important oxidative stress sensor that plays regulatory role in energy metabolism, on metabolic parameters of ovariectomized menopausal model.This study is of interest as it addresses an important biological and medical issue. On the whole, the manuscript is sound and well written but I however have some minor concerns that should be addressed by the authors before publication.

1. Abstract: please include p-value after differences affirmations through the abstract.

We have added the P-value in the abstract.

Materials & methods

2. The authors took in account the the National Institutes of Health guide for the care and use of Laboratory animals? Please, provide this information

We have added the information.

3. Statistical analysis: in the present study the authors compared four groups, T-test seems to not be the best way to perform this analysis. ANOVA must be applied. Please, review all numbered sections through the manuscript.

Thanks for the suggestions. We have redone all the statistical analysis. Analyses were performed using Graphpad Prism using one-way ANOVA with Bonferroni's Multiple Comparison Test or Kruskal-Wallis test with Dunn's Multiple Comparison Test for ordinary or non-parametric data in four groups, as specified in the figure legends and method section. 

Results.

4. Please, provide food intake/energy intake of the animals. Is it hyperphagia presented in ovariectomized mice?

Thanks for the reviewer to provide a very good point that we can do in future. Regretfully, we did not record the food intake of each mice, therefore did not know whether there is hyperphagia presented in the ovariectomzed mice. We would like to explore this in following research for elucidating the detailed mechanisms.

5. Time-varying maps of metabolic indicators in different groups of animals: what the author’s means with “various metabolic substances”? Unclear… please rephrase.

These should be serum LDL, HDL, TG, T-CHO, and GLU, we deleted these unclear phases.

Materials & method.

6.Centrifugation speed should be expressed as g instead of rpm as rpm depends on the diameter of the rotor. Of the centrifuge. Table is available on the web to do this conversion using the speed in rpm and the diameter of the rotor.

We have changed rpm to g.

7. Materials & methods: Please specify the sensitivity of the assay (i.e. the LOD) and the intra assay coefficient of variation of all ELISA assay.

We add the quality control parameters for all biochemical and ELISA kits, including range, sensitivity, intra and inter CV, and minimal sample volumn for all assays, as a supplementary table.

Results:

8.Were the main white adipose tissue fat pads dissected out to evaluate the adiposity of the animals. Does a greater weight gain in the groups also mean fat accretion?

Thanks for the suggestions. As literature demonstrated that Nrf2 deletion inhibits lipid synthesis, we did not think about that the weight gain is due to fat accretion; therefore did not conduct these adiposity analysis in the initial experimental design. However, this could be useful for our further research. 

9. If the adipose tissue accretion is reduced in some groups (here we need food intake to discuss) there is an “energetic paradox” according to the first law of thermodynamics… (Energy intake = Energy expenditure + Energy stored)

Thanks for the suggestions, we would like to do the related experiments in next step.

10. Nuclear factor E2-related factor 2 (Nrf2) is a transcription factor that coordinates the basal and stress-inducible activation of a vast array of cytoprotective genes. Understanding the regulation of Nrf2 activity and downstream pathways has major implications for human health, include in menopause woman. In the presented work, the authors showed the impact of NRF2 depletion on oxidative stress but not in antioxidant response. How this could limit the conclusion of the work?

Thanks for the suggestions. Antioxidant response is believed to play regulatory roles in energy metabolism. These experiments may be conducted to analysis related enzymes such as SOD and NQO1 in the KO menopausal mice. However, based on current data, we did not know how the loss of antioxidant response in the KO menopausal mice affect their energy metabolism. We believed this could be a target to further research to elucidate the detailed mechanisms.

11. Also in discussion section, the evidence of NRF2 and energy metabolism should be clearer.

 We add more discussion on Nrf2 and energy metabolism.

---

## [Decision Letter · Decision Letter 1]

21 Jan 2020

NRF2 deficiency increases obesity susceptibility in a mouse menopausal model

PONE-D-19-25429R1

Dear Dr. Yu,

We are pleased to inform you that your manuscript has been judged scientifically suitable for publication and will be formally accepted for publication once it complies with all outstanding technical requirements.

With kind regards,

Vanessa Souza-Mello, Ph.D.

Academic Editor

PLOS ONE

Additional Editor Comments (optional):

Reviewers' comments:

Reviewer's Responses to Questions

**Comments to the Author**

1. If the authors have adequately addressed your comments raised in a previous round of review and you feel that this manuscript is now acceptable for publication, you may indicate that here to bypass the “Comments to the Author” section, enter your conflict of interest statement in the “Confidential to Editor” section, and submit your "Accept" recommendation.

Reviewer #2: All comments have been addressed

2. Is the manuscript technically sound, and do the data support the conclusions?

Reviewer #2: Yes

3. Has the statistical analysis been performed appropriately and rigorously? 

Reviewer #2: Yes

4. Have the authors made all data underlying the findings in their manuscript fully available?

Reviewer #2: Yes

5. Is the manuscript presented in an intelligible fashion and written in standard English?

Reviewer #2: Yes

6. Review Comments to the Author

Reviewer #2: The authors adequatelly adressed my concern. And in my opinion the manuscript is now acceptable for publication in PlosOne.

7. PLOS authors have the option to publish the peer review history of their article (what does this mean?). If published, this will include your full peer review and any attached files.

Reviewer #2: Yes: Julio Beltrame Daleprane

---

## [Editor Report · Acceptance letter]

29 Jan 2020

PONE-D-19-25429R1 

NRF2 deficiency increases obesity susceptibility in a mouse menopausal model 

Dear Dr. Yu:

I am pleased to inform you that your manuscript has been deemed suitable for publication in PLOS ONE. Congratulations! Your manuscript is now with our production department. 

With kind regards,

on behalf of

Dr. Vanessa Souza-Mello 

Academic Editor

PLOS ONE